# Relationship between Accuracy, Speed, and Consistency in a Modern Pentathlon Shooting Event

**Jongchul Park [1,2]** , **Seunghun Lee [3,*]** and **Sabin Chun [4,*]**

1   Department of Marine Sports, Pukyong National University, Busan 48513, Korea
2   Marine Designeering Education Research Group, Pukyong National University, Busan 48513, Korea
3   Department of Sports Coaching, Catholic Kwandong University, Gangneung-si 25601, Korea
4   Department of Physical Education, Graduate School, Pukyong National University, Busan 48513, Korea
*   Correspondence: lsh8383@naver.com (S.L.); 1004sabin0534@gmail.com (S.C.)

**Abstract:** In the modern pentathlon, the laser run is the highest influence on the results, and fast and accurate shooting is essential. This study evaluates the relationship between shooting characteristics (shot speed, consistency, accuracy) according to sex, competition type, and round number during laser run competitions among 19,648 modern shooting round cases from 2015 to 2019. As a result, men showed faster shot speed than women ($p < 0.05$, $d = 0.493$), and final rounds were significantly better than qualifier round in shot speed ($p < 0.05$, $d = 0.176$), consistency ($p < 0.05$, $d = 0.138$), and accuracy ($p < 0.05$, $d = 0.151$). Series-1 had the highest consistency ($p < 0.05$, $\eta_p^2 = 0.01$) and accuracy ($p < 0.05$, $\eta_p^2 = 0.07$) compared to other series. Series-4 had the lowest shooting speed ($p < 0.05$, $\eta_p^2 = 0.01$) compared to other series. Consistency and speed showed moderate positive correlation ($r = 0.362$, $p < 0.05$). These results show differences in shot characteristics according to sex, competition type, and round number, and explain the relationship between shot speed and consistency. We believe these results will help relevant coaches and players aiming to improve performance understand shot characteristics and reduce shot time.

**Keywords:** running; shooting; modern global pentathletes; sports performance

## 1. Introduction

The modern pentathlon is a multisport competition comprising five different components: fencing, swimming, horseback riding, and laser running (shooting and cross-country running) [1]. The laser run is the last of these five events. During an 800 m cross-country run, the athletes must hit an electronic target that is 10 m away five times. Thereafter, they can continue running. The winner of this event is determined after four rounds of running and hitting the target. The athlete does not incur a penalty if they fail; however, reloading the gun requires time (approximately 3.35 s and 3.60 s for men and women, respectively), which may affect the final competition outcomes [2].

Shooting performance is a fundamental contributor to good overall performance in the modern pentathlon [3]. Slow shooting can improve accuracy and minimize mistakes; however, it increases the total time required to hit the target five times. Contrastingly, shooting the target rapidly decreases the total shooting time, while increasing the chance of mistakes. Ideally, all motions must be performed with speed and accuracy. However, focusing on the shot speed leads to reduced accuracy and vice versa [4]. Maximum accuracy in human motor performance is typically attained at speeds that are slower than the maximum achievable speed [5–7]. Thus, in the modern pentathlon, an optimal balance between the shot speed and accuracy must be established to minimize the total shooting time [8]. In other words, athletes must choose between either shooting quickly with less accuracy or shooting slowly with greater accuracy. Previous studies have evaluated the effects of the pistol movement and body sway on shot accuracy [9,10]. However, in modern

pentathlons, the target sizes and shooting methods vary. Thus, the individual factors that significantly affect the shot accuracy must be identified [11].

In the human motor system, the distinct types of movement do not change; however, the motor speed, distance, and time can differ. This is referred to as the response variability [4,12]. The response variability affects the motor performance accuracy, and the level of variability differs according to the type and characteristic of the given task [13]. During exercise, the response variability is relatively low when the reproducibility is increased through the repetitive performance of a given movement [14]. In particular, a consistent posture while engaging in shooting sports, such as golf, archery, and shooting, is an important determinant of performance [15–18].

The relationship between accuracy and consistency has been proposed as an important contributing factor to improved performance when exercising [19]. Biathlon athletes show repetitive posture control patterns for consistent shooting [20]. It has been shown that elite pentathletes display better performance during better shooting accuracy [3]. Studies on the relationship between golf swing consistency and overall performance [21,22] demonstrated that consistent movement during exercise leads to better performance. In contrast, it is argued that there is little evidence for the association between athletes' specific patterns of behavior and improved performance [23]. Studies that explored the relationship between consistency and performance among elite golf players [24], a study on the relationship between performance and posture consistency targeting high-level archers [16], and a study that confirmed the relationship between the duration of competition and shooting accuracy of pentathlon athletes [25] did not report a correlation between consistent motion and performance. However, these studies had small sample sizes; therefore, their statistical validity could not be confirmed. Previous studies on the importance of consistent movement in sports report contradictory findings. Moreover, for various reasons, several previous studies failed to demonstrate a clear relationship between consistent motion and sports performance [26].

In a modern pentathlon, physical fatigue accumulates during the cross-country running, and athletes must shoot the target while feeling unstable [27]. In addition, time must be spent efficiently; consequently, the accuracy and shooting speed must be compromised. This is a key component that differentiates this event from other shooting events (archery and shooting) that have time restrictions. The shot consistency is thought to contribute to improved performance (reduced shooting time) by minimizing external threats. In contrast, consistency in motion from extensive practice may not reflect the changing external environment (e.g., the physical conditions related to maximum fatigue, the competitive environment, and psychological pressure), and this could lead to poor performances. Thus, to gain an understanding of the optimal methods required to reduce the shooting time, the relationship between the motion consistency and accuracy in the shooting component of modern pentathlons should be investigated. A consistent shooting motion can be inferred from consistent shooting times. The specific amount of shooting time required by top-level athletes when participating in world championship events suggests that these athletes perform consistent motions.

Therefore, beginning in 2015, when official shooting records were first provided, this study analyzed the official world records of the Union International de Pentathlon Modern (UIPM) to investigate the relationship between consistent shot speed, consistency, and accuracy.

## 2. Materials and Methods

### 2.1. Research Data Sources

In this study, data for every match were obtained from the UIPM official website (https://www.uipmworld.org/events/pentathlon-tetrathlon?page=1/, accessed 10 July 2021). The competitions selected for analysis are the World Cup and World Championships, which are prestigious competitions in which top modern pentathlon athletes participate. The qualifiers and main event records of all players participating in the World Cup and

World Championships between 2015 and 2019 were collected. Of a total of 1178 players, 637 players' (347 men and 290 women) data were included for analysis. The exclusion criteria are as follows: (1) unable to verify data, and (2) abstention. In total, the data from 19,648 shooting round cases were divided according to sex (men: 10,221 and women: 9427), type of competition (qualification: 12,318 and final: 7330), and the number of rounds (four series: 4912, respectively).

### 2.2. Variable Selection

UIPM discloses the game records (shooting time, number of snipers) of players in every game, and this study obtained a sample from these data. All the obtained data were processed as follows to calculate shot characteristics (speed, consistency, and accuracy).

First, the shot speed was determined by calculating the mean time (in seconds) required to shoot one shot for each round. A shorter time indicated a faster shot speed.

Second, the shot consistency was assessed using the coefficient of variation (CV), which was the standard deviation of the shot time for each round divided by the average. The mean shooting time could be used to assess the athletes' consistency. The key aspect was the difference in the time between the different shots. A greater difference in the shooting time indicated poor consistency. The consistency of the motions was evaluated using the CV. A lower CV indicated greater consistency and vice versa.

Third, the shot accuracy was assessed as the percentage of shots that hit the target compared to the total shots required for each round. A higher percentage indicated higher accuracy.

Then, the variables were expressed using averages or ratios to evaluate the relationship between shot speed, consistency, and accuracy.

### 2.3. Statistical Methods

After we performed the data cleaning and pre-processing using Microsoft Excel (Microsoft, Redmond, WA, USA), SPSS Statistics (Windows version 22.0; IBM Corp., Armonk, NY, USA) was used to perform the data processing. First, independent sample $t$-tests and one-way analysis of variance (ANOVA) with a 95% confidence interval (CI) were conducted to determine the significant differences in the shot characteristics (speed, consistency, and accuracy) according to sex, competition type, and round number. Scheffe's post-hoc test was used to compare the differences between the significant variables. The Partial eta squared ($\eta_p^2$) or Cohen's d were used to obtain the effect sizes; the effect size magnitudes were interpreted as small (0.01–0.59 and 0.20–0.49, respectively), moderate (0.06–0.139 and 0.50–0.79, respectively), and large ($\geq 0.14$ and $\geq 0.80$, respectively). Second, Pearson's product-moment correlation coefficient ($r$) was used to assess the correlation between the shot characteristics when compared according to sex, competition type, and round number. The level of correlation was interpreted as weak (0.1–0.3), moderate (0.3–0.5), and strong (0.5–1.0). The level of significance was set at $\alpha = 0.05$.

## 3. Results

### 3.1. Analysis of the Group Comparisons

Tables 1–3 show the number of cases, averages, standard deviations, and statistical test results for the shot characteristics according to sex, competition type, and round number.

The independent sample $t$-test was conducted to identify the difference in the shot characteristics of their sex and competition type. As a result, men showed significantly higher shot speed than women ($p < 0.01$, $d = 0.493$); however, there were no statistically significant differences noted in the shot consistency and accuracy. In the competition type, it was confirmed that the final rounds were significantly faster and higher than qualifying in shot speed ($p < 0.01$, $d = 0.176$), consistency ($p < 0.01$, $d = 0.138$), and accuracy ($p < 0.01$, $d = 0151$).

**Table 1.** Differences in the shot characteristics according to sex.

| Variable (Unit) | Groups | N | Mean | SD | $t$ | $p$ | $d$ | 95% CI for $d$ | |
|---|---|---|---|---|---|---|---|---|---|
| | | | | | | | | Lower | Upper |
| Speed (seconds) | Men | 10,221 | 2.274 | 0.343 | 34.551 * | 0.001 | 0.493 | 0.465 | 0.521 |
| | Women | 9427 | 2.451 | 0.376 | | | | | |
| Consistency (CV) | Men | 10,221 | 0.113 | 0.111 | 0.825 | 0.410 | 0.011 | −0.016 | 0.039 |
| | Women | 9427 | 0.111 | 0.106 | | | | | |
| Accuracy (ratio %) | Men | 10,221 | 0.715 | 0.178 | 0.797 | 0.426 | 0.011 | −0.016 | 0.039 |
| | Women | 9427 | 0.713 | 0.176 | | | | | |

\* $p < 0.05$. Abbreviations: CV, coefficient of variation; SD, standard deviation.

**Table 2.** Differences in the shot characteristics according to the competition groups.

| Variable (Unit) | Groups | N | Mean | SD | $t$ | $p$ | $d$ | 95% CI for $d$ | |
|---|---|---|---|---|---|---|---|---|---|
| | | | | | | | | Lower | Upper |
| Speed (seconds) | Qualification | 12,318 | 2.382 | 0.392 | 11.932 * | 0.001 | 0.176 | 0.147 | 0.205 |
| | Final | 7330 | 2.320 | 0.325 | | | | | |
| Consistency (CV) | Qualification | 12,318 | 0.117 | 0.113 | 9.385 * | 0.001 | 0.138 | 0.109 | 0.167 |
| | Final | 7330 | 0.103 | 0.100 | | | | | |
| Accuracy (ratio %) | Qualification | 12,318 | 0.704 | 0.180 | 10.262 * | 0.001 | 0.151 | 0.122 | 0.180 |
| | Final | 7330 | 0.730 | 0.172 | | | | | |

\* $p < 0.05$. Abbreviations: CV, coefficient of variation; SD, standard deviation.

**Table 3.** Differences in the shot characteristics according to the repetition groups.

| Variable (Unit) | Groups | N | Mean | SD | 95% CI | | $F$ | Post Hoc | $\eta_p^2$ |
|---|---|---|---|---|---|---|---|---|---|
| | | | | | Lower | Upper | | | |
| Speed (seconds) | Series-1 | 4912 | 2.354 | 0.368 | 2.344 | 2.365 | 5.873 * | 3, 1, 2 < 4 | 0.01 |
| | Series-2 | 4912 | 2.358 | 0.368 | 2.348 | 2.369 | | | |
| | Series-3 | 4912 | 2.346 | 0.357 | 2.336 | 2.356 | | | |
| | Series-4 | 4912 | 2.376 | 0.386 | 2.366 | 2.387 | | | |
| Consistency (CV) | Series-1 | 4912 | 0.107 | 0.102 | 0.104 | 0.110 | 6.225 * | 1 < 2, 3, 4 | 0.01 |
| | Series-2 | 4912 | 0.111 | 0.104 | 0.109 | 0.114 | | | |
| | Series-3 | 4912 | 0.113 | 0.111 | 0.109 | 0.116 | | | |
| | Series-4 | 4912 | 0.117 | 0.118 | 0.113 | 0.120 | | | |
| Accuracy (ratio %) | Series-1 | 4912 | 0.739 | 0.176 | 0.733 | 0.744 | 45.112 * | 1 > 3, 2, 4 | 0.07 |
| | Series-2 | 4912 | 0.706 | 0.174 | 0.701 | 0.710 | | | |
| | Series-3 | 4912 | 0.707 | 0.176 | 0.702 | 0.712 | | | |
| | Series-4 | 4912 | 0.703 | 0.180 | 0.698 | 0.708 | | | |

\* $p < 0.05$, 1: Series-1, 2: Series-2, 3: Series-3, 4: Series-4. Abbreviations: CV, coefficient of variation; SD, standard deviation.

A one-way ANOVA was conducted to assess the statistical significance of each round sequence. Series-4 was significantly slower than the other series in shot speed ($p < 0.05$, $\eta_p^2 = 0.01$). In consistency and accuracy, series-1 was significantly higher than other series ($p < 0.05$, $\eta_p^2 = 0.01$, $p < 0.05$, $\eta_p^2 = 0.07$, respectively).

### 3.2. Variable Correlation Analysis

Table 4 shows the correlation analysis results for the modern pentathlon shot characteristics. The correlation coefficient for the comparison between the shot consistency

and speed was $r = 0.362$ ($p < 0.01$), demonstrating a moderate positive, linear relationship. The shot accuracy had a weak negative linear relationship to both the shot speed and consistency ($r = -0.018$, $p < 0.05$ and $r = -0.117$, $p < 0.01$, respectively). In terms of the correlation between the groups' shot characteristics, the shot consistency and speed showed a moderate positive correlation in all areas. Additionally, the shot accuracy and speed demonstrated a negative correlation with both the men and the qualifying group. There were no significant correlations noted in relation to the round sequences. Finally, the shot accuracy and consistency showed a weak negative correlation in all areas.

**Table 4.** Correlation of the shot characteristics (shot speed, consistency, accuracy) according to the groups.

| Group | | Consistency and Speed | Speed and Accuracy | Consistency and Accuracy |
|---|---|---|---|---|
| | | *r* | *r* | *r* |
| Total | | 0.362 ** | −0.018 * | −0.117 ** |
| Sex | Men | 0.387 ** | −0.033 ** | −0.108 ** |
| | Women | 0.363 ** | −0.001 | −0.127 ** |
| Competition type | Qualification | 0.389 ** | −0.027 ** | −0.118 ** |
| | Final | 0.289 ** | 0.019 | −0.103 ** |
| Round number | Series-1 | 0.336 ** | 0.002 | −0.091 ** |
| | Series-2 | 0.314 ** | −0.012 | −0.106 ** |
| | Series-3 | 0.387 ** | 0.003 | −0.087 ** |
| | Series-4 | 0.402 ** | −0.058 | −0.169 ** |

* $p < 0.05$, ** $p < 0.01$.

## 4. Discussion

This study examines the differences between shot speed, consistency, and accuracy according to sex, competition type, and round number in order to optimize shooting time. The relationship between the shot speed, consistency, and accuracy was investigated. As a result, men showed higher shot speed than women, and there was no sex difference in accuracy and consistency. In the final, the players tended to have higher shooting speed, consistency, and accuracy than in the qualification. However, the shot speed for series-4 was decreased than in the other series, and the consistency and accuracy were confirmed to be the most consistent and accurate in series-1. Finally, speed and accuracy, and consistency and accuracy showed negligible negative correlations; however, we established a moderate positive correlation between consistency and speed.

Athletes are exposed to a variety of psychological and physiological stressors, such as high physical loads during competition and competing rivals. When high performance is required, it is necessary to quickly overcome these stresses and face the situation, especially when accuracy and speed are required, such as shooting. Here, sports resilience is defined as the ability to positively adapt to stress caused by psychological and physical loads in competitive conditions and factors that reduce negative factors [28–30]. Based on some studies that speculate that there is a positive correlation between psychological resilience and optimal sports, a study was conducted to classify the level of sports resilience according to sport, gender, age, and level of competition [28]. As a result, it was confirmed that there was a difference in the level of sports resilience according to gender and age [28]. Among them, looking at the results according to gender, suggest that men have a better ability to adapt positively in competitive environments than women [28]. In this study also, men did not show any significant difference in shot accuracy and consistency when compared with women, but it was confirmed that they were significantly faster in shot speed. This could mean that women need a little more time than men to achieve similar levels of shooting accuracy and consistency as men. In addition, since the performance in laser runs is closely related to game time, accurate and fast shot speed can help to achieve better results [3].

Thus, these results suggest that men have a better ability to appropriately control external factors when shooting after running than women.

As a result of checking the difference in shooting characteristics according to the type of competition in this study, it was confirmed that all shooting characteristics were improved in the final, not the preliminary. This result is similar to the case where athletes refresh their records more frequently in the finals than in the qualifying matches [28]. It may be a result of the importance of the conditions faced by the players, pressure, appropriate stress, and physiological response [28]. Thus, these results can be considered that psychological and physiological characteristics can be reflected in shooting characteristics depending on the type of competition [27]. In addition, considering that advancing to the finals increases the chances of winning, 0.01 s of a laser run may not be a meaningless number. This study suggested the standard of shot speed (2.320 s), which is generally required in the finals, through the five-year cumulative data provided by UIPM [17]. This criterion is the result of considering the tremor of the muscles due to physiological effects after cross-country, heart and breathing control, along with the previous study that the aiming time of about 2 s is appropriate for the skilled person when shooting a pistol [31]. It is judged that this information can be presented as a limited time to improve accuracy and consistency along with shot speed during laser run training. In conclusion, because the modern pentathlon judges whether the final exists by adding up the scores of the five events, athletes are affected by all other sports to advance to the final [20]. However, in this study, the preliminaries and finals of the laser run showed significant differences in shot speed, accuracy, and consistency, highlighting that these three characteristics are one of the factors contributing to the final participation in the laser run.

The shooting at laser run event occurs during cross-country running. The increase in the round number is directly related to physical factors such as increased heart rate and fatigue accumulation [17,27,32]. Therefore, athletes are inevitably affected by physical strength proportional to the game time when shooting, which may affect their shooting performance. In fact, Bao et al. [33] confirmed, during shooting after 1000 m running, significant differences were noted in the body sway and shot accuracy compared to before running. This demonstrated that, in modern pentathlons, the physiological tremor and physical load of the upper extremity affected the shot accuracy [33]. Additionally, in this study, series-1 did not show a significant difference in shot speed from the other series, but it was confirmed that the accuracy and consistency were significantly high. On the other hand, series-4 showed no significant difference from other series in accuracy and consistency but was found to be significantly slower in shot speed. These results show that in series-1, where the physical load is relatively low, there could be accurate shots without undulation between shots at the optimal speed. However, where the physical load is accumulated such as in series-4, the shot speed may be delayed in order to maintain a similar level of accuracy in other rounds. On the other hand, according to a previous study by Sadowska et al. [27] that analyzed changes in posture balance according to physical load involving 25 pentathletes, the physical load caused by laser run running affects the stability of the athlete during shooting. However, balance disturbances occurring after the first 800m run remain at the same level during shooting after the second 800m, third 800m, and fourth 800m runs, which explained that the level of fatigue does not affect postural balance disturbances in the shooting position [27]. In addition, in a study by Le Meur et al. [34] that confirmed the physiological needs of elite pentathlon athletes during laser running, there was no significant difference in shot success rate and time according to the accumulation of physical load compared with other rounds. Additionally, no change was observed in the pistol and body movements [32]. In this study, there was a significant difference between rounds in shooting accuracy and consistency between series-1 and series-2, -3, and -4, while no significant difference was seen between series-2, -3, and -4. These results show that, although physical loads from running can affect shooting performance, this can be the result of adapting to the additional physical load [33,34]. Athletes have been trained for a long time to compete in the best conditions in all modern pentathlon events: swimming,

fencing, horseback riding, running, and shooting. Therefore, the function of these players suggests that they could keep the shot performance parameters constant to some extent even when the physiological and physical loads continue to increase [27,32–35].

Shot accuracy and speed, and shot accuracy and consistency showed negligible negative correlations; however, shot speed and consistency exhibited moderate positive correlations. These results suggested that the shot speed tended to increase as the shot consistency improved. Several previous studies have assessed the relationship between shot speed or time and accuracy; however, the relationship between shot consistency and accuracy has not yet been evaluated. Specifically, a higher correlation between shot consistency and accuracy was observed in women compared to men, in the qualifying events compared to the final events, and as the rounds progressed. However, the correlation coefficient was low, limiting the statistical significance of the results. Moreover, the correlation coefficient does not indicate that changes in one variable necessarily lead to changes in the other variables. Furthermore, this study was conducted using match records. Thus, the data were not collected under controlled experimental conditions, and possible non-linear relationships were not considered. These limitations suggest that other variables that were not assessed in this study may have affected the correlation between shot speed, accuracy, and consistency. Therefore, we propose that future studies should assess the other possible factors that may have influenced these factors.

In a modern pentathlon, the athletes' performance and results in competitions often depend on the success or failure of a single shot. Therefore any factors that can reduce the time spent shooting by even 0.1 s should be identified. Thus, this study aimed to investigate the relationship between shot speed, consistency, and accuracy in modern pentathlons and was the first study to use the long-term big data of the best players in the modern pentathlon for analysis. However, in this study, only limited data provided by the UIPM official website was used for analysis, and possible nonlinear relationships were not considered. In addition, although the competitions selected for analysis are competitions in which world-class athletes participate, since the classification according to the game level was not considered, the difference in the shooting performance according to the performance level may have been reflected. Additionally, psychological (evaluation of other players) and physical characteristics that could affect the shot characteristics were not considered [27]. Future studies, to identify factors that can affect shot speed, clearly distinguishes the performance level, and the gaze movement of an athlete can be considered during shooting through pupil tracking using an eye tracker [36]. Additionally, heart rate can be used to account for the internal load from running.

### 5. Conclusions

During laser run shooting, men showed a faster shooting speed than women, and the competition type analysis showed a faster shooting speed in the finals than in the qualifying rounds, with high consistency and accuracy. Additionally, in round numbers, series-1 had the highest consistency and accuracy, and series-4 had the lowest shooting speed. Shot speed and accuracy, and consistency and accuracy showed negligible negative correlations, however, consistency and speed showed a moderate positive correlation. The results of this study show that there are differences in shooting characteristics between sex, competition type, and round numbers, and explain the relationship between shot speed and consistency. We believe that these results will help modern pentathlon players understand the shooting characteristics for reducing shooting speed in laser run competitions and provide data for record improvement.

**Author Contributions:** Conceptualization, J.P. and S.L.; methodology, J.P., S.L. and S.C.; software, J.P.; validation, J.P.; formal analysis, J.P. and S.C.; investigation, J.P. and S.L.; data curation, J.P.; writing—original draft preparation, J.P., S.L. and S.C.; writing—review and editing, J.P. and S.L.; supervision, J.P. and S.L.; project administration, J.P. and S.C. All authors have read and agreed to the published version of the manuscript.

**Funding:** This research received no external funding.

**Institutional Review Board Statement:** Analysis of non-human anonymized data.

**Informed Consent Statement:** Not applicable.

**Data Availability Statement:** This research used public data for analysis. https://www.uipmworld.org/events/pentathlon-tetrathlon?page=1 (accessed on 10 July 21).

**Conflicts of Interest:** The authors declare no conflict of interest.

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
