# Peer review of "Relationship between Accuracy, Speed, and Consistency in a Modern Pentathlon Shooting Event"

_applsci, doi:10.3390/app12157852_

Round 1
Reviewer 1 Report
The proposed paper for a publication tackles important question of relationship between Accuracy, Speed and Consistency in a modern pentathlon shooting event. The paper is based prevalently on statistic reports.
From scientific point of view presented paper has many shortcomings which should be improved.
Introduction should be more targeted and improved with research on shooting in connection to the endurance event. Behind motor performances in modern pentathlon there are also many other factors influencing shooting accuracy. With similar problems in shooting accuracy are also dealing biathlon athletes where accuracy, speed and consistency of shooting are very important. There are many research and references in that sport that give deeper insights into shooting accuracy. The suggestion for the authors is to study that references and rewrite completely Introduction and Discussion part of the paper and try to find physiological and psychological insights of their results and not be oriented only into motor behaviour.
The presented paper should be and could be very improved.
Author Response
Thank you for reviewing our thesis
As suggested in the email, I used the 'Track Changes' function. Note please.
Please see the attachment thank you.

Reviewer 2 Report
Relationship Between Accuracy, Speed, and Consistency in a Modern Pentathlon Shooting Event
This is an interesting study in modern pentathlon. I comment the Authors for the research question. Bellow, I provide my major concerns and my point by point comments and suggestions.
Major Concerns:
1. What new does this study provides to modern pentathlon? Authors should highlight the main findings in the discussion and compare them with other studies and with real world situations.
2. Research data sources should be clearly presented for readers. With that said Authors are requested to add more details concerning the website address of UIPM and attached a supplementary file from the online data search.
3. What was the level of the athletes participated in the current study? Was the age of the athletes or their training experience affecting the results? This should be discussed.
4. Units of measurement are missing throughout the manuscript and abstract (not in tables).
5. How the variables were calculated? Authors should clearly present the method they used to measure shot-time, CV% and shot accuracy.
Minor comments:
Abstract:
Line 16: Please add measuring units.
Add statistical indexes such as p values and r-Pearson correlation coefficient.
Line 22-24: Provide a clear message to readers. These lines are confusing and not helping with the study outcomes.
Introduction:
Introduction is well-written and provides a good overview of the literature. Authors present the modern pentathlon and even a reader who does not know what modern pentathlon is, can fully understand it. Good job here, nothing to mention.
Line 93: Make sure that this number (1178 participant) is compatible with the number given at participants section (637).
Methods:
The study would greatly benefit from the addition of a supplementary file of the raw data as well as from the actual website address of UIPM. Also, Authors should provide a small paragraph with the training experience or the biological age of the participants. If Authors manage to find anthropometric characteristics, then this will enhance the quality of the manuscript.
Lines 105-112: These lines are redundant. I am not sure why are here.
Line 115: Provide more details about the measurements. For example, where the ratio is referred to?
Line 116-126: How the variables were calculated? Was there a video from each athlete? Please, clearly present how the variables were collected.
I suggest to Authors adding an effect size for the comparisons (i.e. Cohen’s d, Eta Squared, Hedges g) and include it inside the tables 1, 2 and 3.
Results:
Please make sure that all numbers inside the text have a measuring unit.
Tables are great. Well-done. But, there seems to be a confusion with the N of participants. There are three different numbers presented in the manuscript: 1178 in the abstract, 637 in the participants and 10218 males + 9425 females in table 1 and goes on in Table 2 and 3. Authors should explain why this is happening in Methods before presenting the results.
Line 178-179: I am not sure why figure is presented. If Authors want to use the figure in the discussion section to support a finding then leave it. If not, I suggest deleting it.
Discussion:
First three lines of discussion are confusing. Was is a comparison or a correlational study? I suggest Authors present the purpose of the study and then proceed point by point to presenting their thoughts.
Lines 197-199: What did Maddalena et al., actually found?
Lines 213-221: The study of Bao et al., investigate the effect of running 1000m or 2000m on a treadmill and then shooting. The way this is written here seems that athletes shoot at 1000 or 2000 m distance. Please, rewrite this part.
Line 220: Also, explain body function recovery as presented by Bao et al.
Lines 247-249: Please present the references at the end of the sentence.
Limitations are normally presented in the penultimate paragraph of discussion. Authors should clearly state the limitations of the study as well as the strong aspects of the study.
Last paragraph: Authors examine differences between variables as well as relationships between variables. Thus, the conclusion paragraph should include both findings.
Author Response
Thank you for reviewing our thesis
As suggested in the email, I used the 'Track Changes' function. Note please.
Please see the attachment thank you.
|
Point raised by referee (please summaries) |
Response by author (briefly explain) |
Location in text: Page and paragraph reference |
|
Please add measuring units. |
Thank you for your comment. The abstract has been completely rewritten to help readers understand.
|
Abstract, lines 11-23 |
|
Add statistical indexes such as p values and r-Pearson correlation coefficient. |
Thanks for the comment, you suggested p-value and effect size or r-Pearson correlation. |
Abstract, lines 15-19 |
|
Provide a clear message to readers. These lines are confusing and not helping with the study outcomes. |
We agree with our reviewers. So, I rewrote the conclusion. |
Abstract, lines 21-23 |
|
Introduction should be more targeted and improved with research on shooting in connection to the endurance event. Behind motor performances in modern pentathlon there are also many other factors influencing shooting accuracy. With similar problems in shooting accuracy are also dealing biathlon athletes where accuracy, speed and consistency of shooting are very important. There are many research and references in that sport that give deeper insights into shooting accuracy. |
We agree with the reviewer's suggestion. We have revised the text accordingly. Previously cited references were reviewed. In addition, the literature related to the subject of this study was newly presented. |
Introduction, lines 56-72 |
|
Make sure that this number (1178 participant) is compatible with the number given at participants section (637). |
We agree with our reviewers. There was insufficient explanation for the change in the number of samples. Therefore, an additional explanation is provided in 'Method'. The number of samples previously presented in the introduction has been deleted to reduce confusion. |
Materials and Methods, lines 100-105
|
|
The study would greatly benefit from the addition of a supplementary file of the raw data as well as from the actual website address of UIPM. |
We agree with our reviewers. Therefore, we have provided a link to the website from which we collected the data. |
Materials and Methods, lines 96-97 |
|
Authors should provide a small paragraph with the training experience or the biological age of the participants. If Authors manage to find anthropometric characteristics, then this will enhance the quality of the manuscript. |
We agree with our reviewers. However, we did not find any athlete anthropometric characteristics. Nevertheless, the competitions selected for analysis in this study are prestigious international competitions involving high-level athletes. Therefore, this study has additionally entered the paper about these facts. |
Materials and Methods, lines 97-99 |
|
These lines are redundant. I am not sure why are here |
Sorry, it has been confirmed that the error was not checked before submission. this part has been deleted |
|
|
Provide more details about the measurements. For example, where the ratio is referred to? |
I agree with the reviewer's opinion. The description of the raw data used in the analysis was lacking. All raw data are provided by the International Modern Pentathlon Association. Raw data includes shooting time and total number of shots for all series. Based on this data, we can find the average shot speed to determine fast and slow (CV). Also, we can calculate the accuracy based on the number of shots. |
Materials and Methods, lines 108-110 |
|
I suggest to Authors adding an effect size for the comparisons (i.e. Cohen’s d, Eta Squared, Hedges g) and include it inside the tables 1, 2 and 3 |
We agree with the reviewer. We have revised the content of Tables 1, 2, and 3 accordingly. |
Tables 1, 2, and 3 |
|
Please make sure that all numbers inside the text have a measuring unit. |
We agree with our reviewers. However, all the methods of explaining the results have been modified. Therefore, there is no need to explain the units. |
Results, lines 141-154 |
|
Tables are great. Well-done. But there seems to be a confusion with the N of participants. There are three different numbers presented in the manuscript: 1178 in the abstract, 637 in the participants and 10218 males + 9425 females in table 1 and goes on in Table 2 and 3. Authors should explain why this is happening in Methods before presenting the results. |
Thanks for the reviewer's comments. We reviewed all data to identify and correct errors, and provided an explanation for the number of samples in the 'method'. |
Materials and Methods, lines 100-105
|
|
I am not sure why figure is presented. If Authors want to use the figure in the discussion section to support a finding, then leave it. If not, I suggest deleting it. |
We agree with our reviewers. So, I deleted the picture. |
|
|
The suggestion for the authors is to study that references and rewrite completely Introduction and Discussion part of the paper and try to find physiological and psychological insights of their results and not be oriented only into motor behaviour. |
We appreciate the comments of our reviewers and we very much agree. Therefore, the discussion was revised overall by reviewing the papers on similar sports related to modern pentathlon shooting. This includes explanations considering psychological and physical factors. thank you |
Discussion, lines 182-256 |
|
First three lines of discussion are confusing. Was is a comparison or a correlational study? I suggest Authors present the purpose of the study and then proceed point by point to presenting their thoughts |
We agree with the reviewers and thank you for pointing out. Based on the reviewer's comments, the content has been revised in general. |
Discussion, lines 172-181 |
|
What did Maddalena et al., actually found? |
What the authors have confirmed is that there is a difference in accuracy and consistency depending on the level of play. This content has been removed as it has resulted in a general revision of the discussion. |
|
|
The study of Bao et al., investigate the effect of running 1000m or 2000m on a treadmill and then shooting. The way this is written here seems that athletes shoot at 1000 or 2000 m distance. Please, rewrite this part. |
We agree with the reviewer's opinion. However, as the discussion was revised in general, the section describing the study of Bao et al., was revised. Nevertheless, reflecting the reviewer's opinion (the first 800m and the second 800m) were used in the section where a similar explanation was required. |
Discussion, lines 226-229 Discussion, lines 240-243 |
|
Also, explain body function recovery as presented by Bao et al. |
The shooting characteristics of the athletes decreased overall after the first run compared to before the run, but no significant difference was found in the shooting characteristics between the second, third, and fourth runs and the first run. Therefore, it explained that shot accuracy can remain constant even as physiological and physical loads increase. |
Discussion, lines 247-251 |
|
Please present the references at the end of the sentence. |
Thanks for the reviewer's advice. Overall, errors in the paper were corrected in the process of reviewing and correcting the content. thank you |
|
|
Limitations are normally presented in the penultimate paragraph of discussion. Authors should clearly state the limitations of the study as well as the strong aspects of the study. |
We agree with our reviewers. Therefore, the advantages, limitations, and limitations of the study are described by moving them from the last to the second paragraph. |
Discussion, lines 273-288 |
|
Authors examine differences between variables as well as relationships between variables. Thus, the conclusion paragraph should include both findings. |
We agree with our reviewers. Thanks for the advice. Reflecting the opinions, two facts were explained in detail to draw conclusions. thank you |
Conclusion, lines 290-300 |
|
|
Additional references are provided. |
References, lines 348-351 References, lines 354-357 References, lines 365-366 References, lines 372-380 References, lines 389-393 |
|
|
|
|

Round 2
Reviewer 1 Report
The paper was improved according to some previous considerations, esspecialy discussion. Paper needs English text editing. For example: Abstract - Line 15 - finals, probably finalist have...
Lines 21-22 Need overall English editing
References should be edited according to instructions.
Author Response
Thank you for reviewing my thesis.
The paper has been revised in consideration of valuable comments.
Please see the attachment.
Thank you.
Manuscript ID: applsci-1796939 (Applied sciences)
Title: Relationship Between Accuracy, Speed, and Consistency in a Modern Pentathlon Shooting Event
|
Point raised by referee (please summaries) |
Response by author (briefly explain) |
Location in text: Page and paragraph reference |
|
Paper needs English text editing. For example: Abstract - Line 15 - finals, probably finalist have... |
I agree with the reviewer's advice. The abstract has been edited to make the meaning clearer. thank you |
Abstract, lines 15-16 |
|
Lines 21-22 Need overall English editing |
I agree with the reviewer's opinion. Sorry for not caring. I have checked it again and amended it to make the meaning clearer. please check. thank you |
Abstract, lines 21-23 |
|
Abstract: Delete or rephrase the lines 21-23. |
I agree with the reviewer's opinion. Edited to convey the meaning the author wants. please check. thanks for the advice |
Abstract, lines 21-23 |
|
References should be edited according to instructions |
Thanks for the reviewer's advice. After reviewing all sections of the thesis, the notation method was revised, and the method of writing references was also completed. Please confirm. thank you |
|
|
|
|
|
Thank you.

Reviewer 2 Report
Abstract: Delete or rephrase the lines 21-23.
Author Response

(The authors gave the same response as above.)
